# Prehospital identification of intracerebral haemorrhage: a scoping review of early clinical features and portable devices

Mohammed Almubayyidh [1,2] Ibrahim Alghamdi,[1,3] Adrian Robert Parry-Jones,[1,4] David Jenkins [5]

For numbered affiliations see end of article.

**Correspondence to**
Mohammed Almubayyidh; mohammed.almubayyidh@postgrad.manchester.ac.uk

## ABSTRACT

**Introduction** Prehospital identification of intracerebral haemorrhage (ICH) in suspected stroke cases may enable the initiation of appropriate treatments and facilitate better-informed transport decisions. This scoping review aims to examine the literature to identify early clinical features and portable devices for the detection of ICH in the prehospital setting.

**Methods** Three databases were searched via Ovid (MEDLINE, EMBASE and CENTRAL) from inception to August 2022 using prespecified search strategies. One reviewer screened all titles, abstracts and full-text articles for eligibility, while a second reviewer independently screened 20% of the literature during each screening stage. Data extracted were tabulated to summarise the key findings.

**Results** A total of 6803 articles were screened for eligibility, of which 22 studies were included for analysis. Among them, 15 studies reported on early clinical features, while 7 considered portable devices. Associations between age, sex and comorbidities with the presence of ICH varied across studies. However, most studies reported that patients with ICH exhibited more severe neurological deficits (n=6) and higher blood pressure levels (n=11) at onset compared with other stroke and non-stroke diagnoses. Four technologies were identified for ICH detection: microwave imaging technology, volumetric impedance phase shift spectroscopy, transcranial ultrasound and electroencephalography. Microwave and ultrasound imaging techniques showed promise in distinguishing ICH from other diagnoses.

**Conclusion** This scoping review has identified potential clinical features for the identification of ICH in suspected stroke patients. However, the considerable heterogeneity among the included studies precludes meta-analysis of available data. Moreover, we have explored portable devices to enhance ICH identification. While these devices have shown promise in detecting ICH, further technological development is required to distinguish between stroke subtypes (ICH vs ischaemic stroke) and non-stroke diagnoses.

## STRENGTHS AND LIMITATIONS OF THIS STUDY

⇒ To the best of our knowledge, this is the first review focused on identifying spontaneous intracerebral haemorrhage in the prehospital setting.

⇒ No time of publication restrictions were applied to comprehensively map the available evidence.

⇒ The review is limited to English-language publications only.

⇒ Consistent with the scoping review methodology, a formal risk of bias assessment was not conducted.

## INTRODUCTION

Prehospital recognition of intracerebral haemorrhage (ICH) among suspected stroke cases is not part of standard care, but the ability to reliably do so may pave the way for much quicker delivery of time-critical treatments and triage to a hospital that can provide appropriate interventions (eg, hyperacute stroke care, neurosurgery).[1 2] The importance of the rapid identification of patients with ICH has been highlighted by studies showing that early haematoma expansion (HE) frequently occurs within 2–3 hours of onset.[2–4] Thus, lowering blood pressure (BP) as well as administering anticoagulant reversal/haemostatic agents may limit HE, thereby increasing the likelihood of favourable outcomes.[2 5] These treatment strategies are currently under investigation in prehospital settings.[6–8] Facilitating the diagnosis of ICH in ambulances through simple and cost-effective methods could support the broader implementation of such therapeutic trials in the future.

While the initial diagnosis of stroke in the prehospital setting is based on stroke recognition tools (eg, FAST—Face Arm Speech Test, CPSS—Cincinnati Prehospital Stroke Scale), these tools have been validated to identify all strokes rather than ICH specifically.[9] Recognising the importance of timely and accurate identification of ICH, the American Heart Association/American Stroke Association's

recent guidelines for the management of patients with ICH have highlighted the need to improve the prehospital recognition of ICH.[9] The recent RACECAT trial further emphasises the significance of prehospital decisions in stroke management, which found that bypassing the closest stroke centre for direct transport to a thrombectomy-capable centre was associated with worse outcomes for patients with ICH.[10] This approach could be more beneficial for patients with large vessel occlusion (LVO),[11] underscoring the necessity for the prehospital differentiation of ICH, especially from LVO.[12]

Several clinical features were found to be predictive of ICH in patients who had a stroke who experience persistent symptoms for more than 24 hours.[13] These features were used to develop clinical prediction rules to distinguish ICH from ischaemic stroke (IS).[13 14] Nevertheless, the diagnostic accuracy of these rules is low, and they may not be applicable to patients presenting within a few hours of the onset of symptoms or during the prehospital phase.[15 16] Such tools for the identification of ICH may have the potential to improve patient outcomes. However, it is unclear which clinical features could be used to identify ICH in prehospital settings. Consequently, brain imaging (CT and/or MRI) is the only accepted method for diagnosing ICH and other subtypes of stroke. This has led to the development of new portable imaging devices for the detection of patients with stroke in prehospital settings.[17 18]

The aim of this study is to perform a scoping review to summarise the early distinguishing clinical features of ICH and evaluate novel portable devices that can help prehospital personnel differentiate ICH from other causes of suspected stroke.

## METHODS

The scoping review followed the guidelines of the Joanna Briggs Institute (JBI).[19 20] It was reported according to the Preferred Reporting Items for Systematic Reviews and Meta-Analyses Extension for Scoping Reviews (PRISMA-ScR) checklist (online supplemental appendix 1).[21] We adopted a scoping review approach because of a broad set of inclusion criteria. A detailed study protocol was published previously.[22]

### Eligibility criteria

The eligibility criteria for this review were classified using the 'Population, Concept, Context' framework suggested by the JBI (table 1).[19] The framework is applied as follows:

► Population: Adult patients (aged ≥16 years) with suspected stroke confirmed by either CT or MRI scans. Studies were excluded if they enrolled children (aged <16 years) or combined the clinical features of patients with spontaneous ICH with those of other conditions (eg, subarachnoid haemorrhage (SAH)) in the same group without reporting them separately.

► Concept: First, to distinguish spontaneous ICH from other stroke and non-stroke diagnoses by investigating the early clinical features observed during the prehospital phase or within the initial 24 hours after the onset of symptoms on arrival at the hospital. Second, to explore the use of portable devices in the detection and classification of ICH. Studies were excluded if they reported clinical features of ICH after 24 hours of symptoms onset, or if they investigated clinical features associated with ICH without comparing them against other suspected stroke cases. Further studies excluded from the review included those that did not provide information on the diagnostic accuracy of the tested technology, examined advances in conventional imaging techniques (CT and MRI) or exclusively tested the technology on phantoms or animal models.

► Context: Prehospital and in-hospital settings with no restrictions on the country of study, ethnicity, gender, or socioeconomic status. Studies conducted in primary care settings were excluded.

### Sources of information and search strategy

We included data from peer-reviewed primary research studies published in English. Conference abstracts, commentaries, surveys, case reports, preprints, animal studies and non-English-language papers were excluded. The literature search of MEDLINE (via Ovid), EMBASE

**Table 1** Inclusion and exclusion criteria

| PCC | Included | Excluded |
|---|---|---|
| Population | ► Studies of adult patients (≥16 years old) with suspected stroke. | ► Studies of children (<16 years old).<br>► Studies combined ICH clinical features with other aetiologies. |
| Concept | ► Studies reported the clinical features of suspected stroke patients, obtained during prehospital care or within 24 hours of symptom onset on hospital admission.<br>► Studies evaluated portable devices to detect ICH. | ► Studies examined the clinical features of patients after 24 hours of symptoms onset or focused on ICH-related clinical features without comparisons to other suspected stroke cases.<br>► Studies that failed to provide information on the diagnostic performance of the technology being tested, or examined advances in conventional detection methods, or restricted their testing to phantoms or animal models. |
| Context | ► Prehospital and in-hospital studies. | ► Primary care studies. |

ICH, intracerebral haemorrhage; PCC, Population, Concept, Context.

(via Ovid) and CENTRAL (via Ovid) databases was conducted in August 2022, using our prespecified search strategies (online supplemental appendix 2).[22] To ensure that all relevant studies were included, there was no restriction regarding the date of publication. In addition, we searched the reference lists of retrieved papers to identify additional relevant studies for this review.

## Study selection

To eliminate duplicates and facilitate screening and collaboration, the identified studies were imported into EndNote and Rayyan Qatar Computing Research Institute (QCRI) software.[23] Two independent reviewers (MA and IA) selected eligible studies in two stages; initially, titles and abstracts were screened, and then full texts were assessed. MA screened all titles, abstracts and full-text articles against the eligibility criteria. IA screened a random sample of 20% of the studies at both the title and abstract and full-text stages. Any disagreements in any phase of the selection process were resolved by discussion between reviewers to reach consensus or by a third reviewer (ARP-J or DJ).

## Data extraction

Two data extraction forms (online supplemental tables 1 and 2) were created using Microsoft Excel, considering the review objectives, the content of the included studies and insights from previous reviews.[15 24] These tables were designed to present the data from the different studies in a standard format, allowing comparisons to be made between studies and their findings. The data were extracted by the first reviewer (MA), and the second reviewer (IA) independently checked 20% of the extracted data to ensure accuracy.

## Data synthesis and quality assessment of studies

Data were summarised as median, mean and range when appropriate. Otherwise, a narrative approach was used to synthesise the findings of the included studies. In addition, since the PRISMA-ScR checklist does not require an assessment of bias,[21] a formal risk of bias assessment was not conducted.

## Patient and public involvement

No patients or public were involved in this study.

## RESULTS

The search strategies yielded a total of 10 490 articles; after removing duplicates, 6803 articles remained (figure 1). Following the initial screening of titles and abstracts, 182 articles remained as potentially relevant and were retrieved for full-text review. After obtaining and reviewing the full texts, 162 studies were excluded with reasons, as illustrated in a PRISMA flow chart in figure 1. The manual search of the reference lists of included studies revealed two additional studies, resulting in a total of 22 included studies.

## Description of included studies

The retrieved studies were divided into two groups based on the aims of the scoping review: studies that reported the early clinical features of patients with ICH (n=15)[25–39] and those that used portable devices to detect ICH (n=7).[40–46] These groups were divided further according to the data extracted from them.

In the first group (online supplemental table 1), studies compared the clinical features among various populations, including stroke subtypes (n=12) and non-stroke diagnoses (n=3). Different statistical analyses were used in these studies to determine clinical features, such as univariable (n=3) and multivariable (n=12) regression analyses. Regardless of the statistical techniques used, the reported clinical features of interest were arranged during the analysis process into three groups (table 2): patients' general information and medical history, early signs and symptoms, and vital signs. Studies that employed portable devices (online supplemental table 2) to detect or differentiate ICH and other diagnoses were divided into four subgroups based on the underpinning technology: microwave imaging technology (n=1), volumetric impedance phase shift spectroscopy (VIPS) (n=1), transcranial ultrasound (n=3) and electroencephalography (EEG) (n=2). Additionally, most studies were conducted in Europe (n=13), followed by North America (n=5) and Asia (n=4).

## Prediction of ICH in suspected stroke patients using early clinical features

### Patients' general information and medical history

Several studies found that patients with ICH were younger (online supplemental table 1),[25–27 33 36–38] and more likely to be male[29 35 36 38] than those with other stroke and non-stroke diagnoses. However, this association was not uniformly reported, as other studies found that neither age[28 29 31 35 39] nor sex,[25–28 31 33 34 39] were predictors of the final diagnosis of ICH. Only one study identified an association between older age and increasing probability of posterior fossa ICH.[34]

Several conditions or factors were identified as being more strongly associated with the presence of ICH (vs other diagnoses), including a history of hypertension (HTN),[30 34 38] prior ICH,[26 31 36] haemodialysis,[26] alcohol consumption[36] and the use of warfarin.[34] Conversely, patients with ICH were less likely to have a history of atrial fibrillation (AF),[26 27 29 34–36 38] coronary artery disease (CAD),[25 26 36 38 39] dyslipidaemia,[26 34 36 37] diabetes mellitus (DM),[25 27] peripheral arteriopathy (PAD),[36] prior transient ischaemic attack (TIA)[33] or IS.[28] However, other studies found no or weak associations between ICH and HTN,[26 27 29 33 35–37] prior TIA/IS,[26 36] history of ICH,[28] AF,[25 33 37 39] CAD,[33–35] dyslipidaemia,[25 35 38 39] DM,[26 31 33–39] PAD,[35] alcohol consumption[35 38] or warfarin use.[28] In two studies,[25 39] a history of HTN was found to be more common among patients who had an IS than patients with ICH.

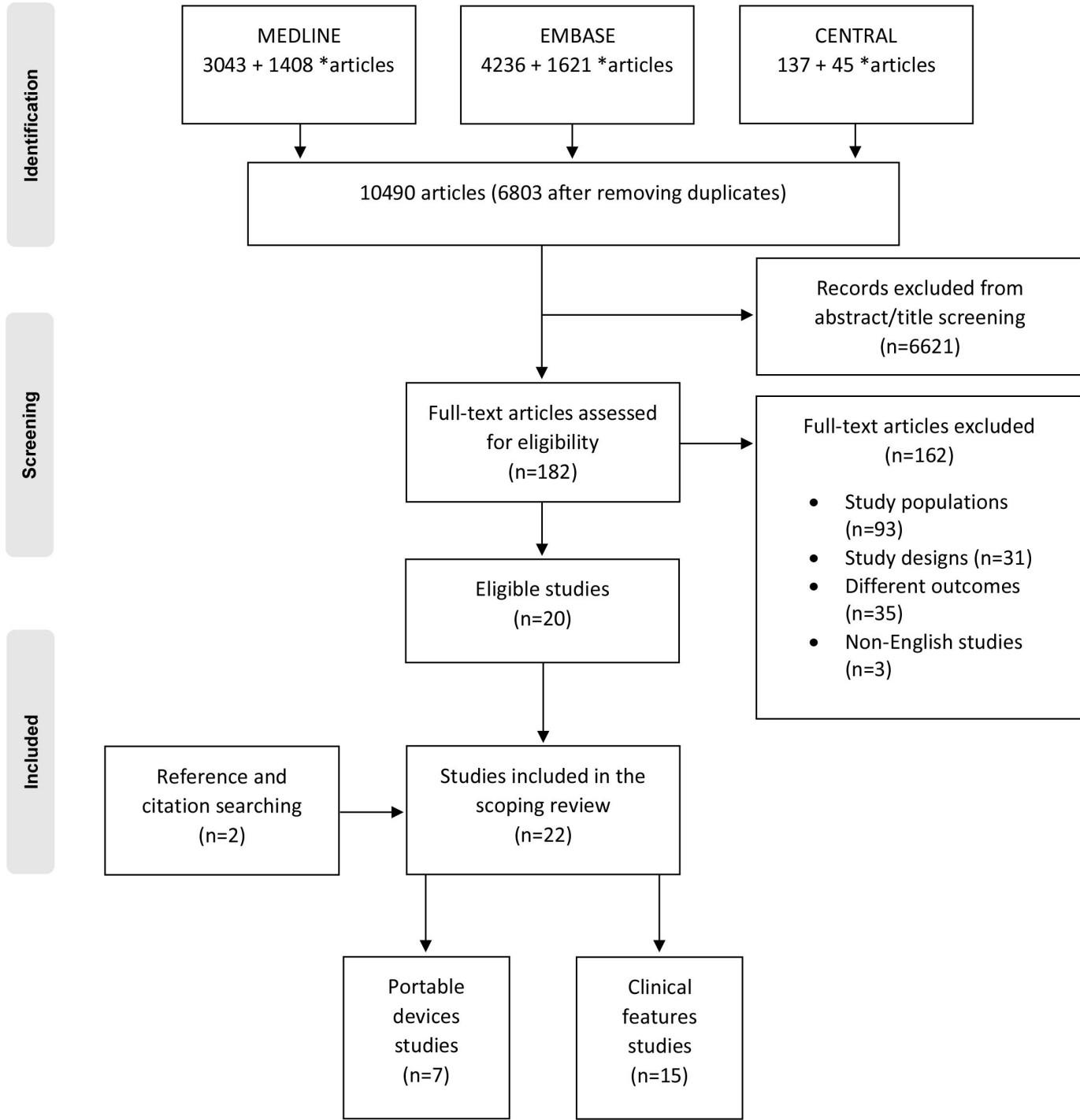

**Figure 1** Flow chart of study selection process (*search results for clinical features+portable devices, respectively).

## Early signs and symptoms of ICH

Several symptoms were reported at the onset of ICH. Among them, headache[26 33 34] and vomiting[26 27 34] were more frequently recorded; however, three studies[25 28 30] did not find these symptoms to be specific to ICH. Additionally, five studies reported that impaired level of consciousness (LOC) was more prevalent among patients with ICH.[25 26 29 32 33] Meanwhile, two studies[28 30] found decreased LOC was not only associated with ICH but also with LVO and SAH.

Seizures[26] and neurological deficits,[26 29] including sensory or motor deficits,[26 29] difficulty in following commands[26 29] and impaired vision[29] were also common presentations. Yet six studies[25 28–30 33 34] did not find the presence of seizures,[25 29 34] or neurological deficits[25 28 30 33] to be specific discriminators of ICH. In terms of neurological deficits, one study found that compared with patients who had an IS, patients with ICH less frequently woke up with symptoms and they experienced greater leg weakness.[33]

**Table 2** Summary of reported clinical features for predicting ICH in suspected stroke patients

| Category | Findings | References |
|---|---|---|
| Patients' general information and medical history | ▶ Age and sex showed inconsistent predictive value for ICH.<br>▶ Hypertension, prior ICH, haemodialysis, alcohol consumption and warfarin use were more frequently associated with ICH.<br>▶ Atrial fibrillation, coronary artery disease, dyslipidaemia, diabetes mellitus, peripheral artery disease and previous TIA/IS were less commonly associated with ICH.<br>▶ Some studies found weak or no associations between the aforementioned factors or conditions and the prediction of ICH. | 25–31 33–39 |
| Signs and symptoms | ▶ Headaches, vomiting, impaired consciousness and seizures were reported, but these symptoms were not specific to ICH.<br>▶ Awakening with stroke symptoms was less likely to be associated with ICH.<br>▶ Patients with ICH exhibited more neurological deficits as measured by the NIHSS. | 25–30 32–39 |
| Vital signs | ▶ Elevated blood pressure levels were more consistently observed at the onset of ICH.<br>▶ Arrhythmias were less frequently seen in patients with ICH.<br>▶ High blood glucose levels were reported in patients with ICH. | 25–29 31–35 37 38 |

ICH, intracerebral haemorrhage; IS, ischaemic stroke; NIHSS, National Institutes of Health Stroke Scale; TIA, transient ischaemic attack.

Eight studies[26 29 34–39] used the National Institutes of Health Stroke Scale (NIHSS) to assess stroke severity either in the prehospital setting[29] or at hospital admission.[26 34–39] Among these studies, six reported higher NIHSS scores at presentation in patients with ICH compared with those who had an IS, with median scores ranging from 11 to 17 for patients with ICH and from 6 to 13 for patients who had an IS.[26 29 35–38] However, two studies found no discernible difference in NIHSS scores between patients with ICH and patients who had an IS.[34 39]

### Vital signs and their predictive capabilities of ICH
Most of the studies investigated vital signs, including systolic and diastolic BP (SBP and DBP, respectively),[25–29 31–35 37 38] mean arterial pressure (MAP),[29 31] heart rate (HR) or rhythm,[25–28 33] peripheral oxygen saturation (SPO$_2$)[33] and blood glucose level.[34 35 38]

In four studies that investigated the prehospital BP,[27–29 33] the BP values were dichotomised as follows: SBP, >200 mm Hg,[33] ≥180 mm Hg,[27 29] ≥165 mm Hg[28]; DBP, ≥110 mm Hg,[29] ≥100 mm Hg,[27] ≥95 mm Hg,[28] >90 mm Hg[33]; MAP, ≥130 mm Hg.[29] These studies found that a higher proportion of patients with ICH had elevated BP at presentation than those with other stroke subtypes and non-stroke conditions.

Six studies reported that mean or median SBP,[26 31 34 35 37 38] DBP[26 31 34 35 38] and MAP[31] values of patients with ICH were higher at the time of onset than in patients with other diagnoses (SBP ranged from 160 mm Hg to 204 mm Hg, DBP ranged from 84 mm Hg to 102 mm Hg and the MAP was 122.0±20.2 mm Hg). Only one study[25] found no differences in the SBP and DBP values of patients with ICH and patients who had an IS in the prehospital setting. However, another study found that elevated SBP (≥200 mm Hg) in the prehospital setting among patients who had a stroke with impaired LOC was strongly associated with an increased risk of ICH.[32]

With the exception of five studies, no differences were detected in vital signs such as HR[27 33] or rhythm,[25 33] SPO$_2$[33] and blood glucose[35] between patients with ICH and patients who had an IS. Three of the five studies reported that arrhythmia was less likely to be associated with ICH,[26–28] and two observed higher median blood glucose levels at the time of ICH onset (7.1–7.4 mmol/L) compared with IS (6.7–6.8 mmol/L).[34 38]

### Portable devices to detect and differentiate ICH
#### Microwave imaging technology
Two pilot studies were conducted to assess the diagnostic accuracy of microwave-based prototype devices in distinguishing ICH from IS, as well as ICH from healthy individuals.[40] In the first study, 20 patients who had a stroke were examined within 7–132 hours of onset, using the first prototype device by engineering and neurophysiology staff. When aimed at identifying all patients with ICH, the haemorrhagic detector correctly differentiated 7 out of 11 patients who had an IS from the ICH group, with an area under curve (AUC) of 0.88.

The second pilot study involved 25 patients who had a stroke within 4–27 hours of onset and 65 healthy individuals, with nursing staff testing the second custom-built helmet. When aimed at identifying all patients with ICH, the haemorrhagic detector accurately distinguished 14 out of 15 patients who had an IS from the ICH group, achieving an AUC of 0.85 for distinguishing ICH from IS and 0.87 for distinguishing patients with ICH from healthy individuals.

#### Volumetric impedance phase shift spectroscopy
The effectiveness of the VIPS visor device, developed by Cerebrotech, was investigated in a study aiming to assess its capability to measure the bioimpedance of each brain hemisphere for determining stroke severity.[41] The study enrolled 248 participants, including 41 'code stroke'

patients, 79 healthy controls and 128 patients with various neurological conditions at a comprehensive stroke centre. Trained personnel conducted three scans using the visor device, with a total scan duration of approximately 30 s.

The study aimed to evaluate the device's performance in distinguishing severe strokes from minor strokes and severe strokes from all other subjects. Severe strokes were defined as those requiring triage to a comprehensive stroke centre, including conditions such as LVO, NIHSS score of ≥6, ICH of ≥60 mL and large territorial strokes.

The results of the study demonstrated that patients who had severe strokes exhibited the highest mean bioimpedance asymmetry. The VIPS visor device achieved a sensitivity of 93% (95% CI 83% to 98%), specificity of 92% (95% CI 75% to 99%) and an AUC of 0.93 (95% CI 0.85 to 0.97) when differentiating severe strokes from minor strokes. Similarly, in distinguishing severe strokes from all tested subjects, the device showed a sensitivity of 93% (95% CI 83% to 98%), a specificity of 87% (95% CI 81% to 92%) and an AUC of 0.93 (95% CI 0.89 to 0.96). The ability of VIPS to differentiate ICH from other conditions was not reported in this study.

## Transcranial ultrasound

Three studies used transcranial ultrasound imaging to diagnose ICH and distinguish it from other suspected stroke causes.[42–44] In one clinical study,[42] transcranial colour-coded duplex sonography (TCCS) scans were conducted on 84 patients using a Sonos 1000 Hewlett-Packard machine. ICH diagnosis was based on sharply demarcated hyperechogenic areas in the brain tissue. Among the patients, 52 had IS or TIA, 15 had ICH, and 17 were excluded due to inadequate temporal bone windows. TCCS identified ICH in 14 patients with focal hyperechogenicity, with only one small haemorrhage missed. Sensitivity and specificity were reported as 88% and 96%, respectively.

In another study,[43] the diagnostic accuracy of TCCS was evaluated using Siemens Sonoline Elegra or Acuson 128 XP/4 on 151 patients with acute hemiparesis. Among them, 60 had ICH, 67 had IS, and 24 had inconclusive CT scans. Ultrasound examination of 18 patients revealed insufficient acoustic bone windows. For the remaining 133 patients, TCCS detected 50 cases of ICH with a sensitivity of 94% and specificity of 95%.

A third study examined 107 suspected stroke patients using SonoSite M-Turbo, Philips Sparq or Philips CX50 for TCCS.[44] Among the patients, 63 had acute IS, 18 had ICH, 13 had TIA, and 13 had stroke-mimicking conditions. Sonographic examination revealed an insufficient temporal bone window in 18 cases. Ultrasound imaging accurately identified ICH in 56% of patients but failed in 33% of cases. The study presented a triage model that combined transcranial ultrasound and clinical assessment, resulting in a 10% improvement in ICH detection compared with clinical assessment alone. While the specificity of both techniques was reported as 99%, the

sensitivity of the combined model was 63%, whereas clinical assessment alone yielded a sensitivity of 6%.

## Electroencephalography

Two studies examined the diagnostic capability of portable EEG devices for detecting strokes in emergency departments (EDs).[45 46] In one study,[45] BrainScope's hand-held device was used to identify strokes in 183 patients presenting with stroke-like symptoms. Among them, 31 patients were diagnosed with IS, 17 with haemorrhagic stroke and 135 with stroke-mimicking conditions. The device recorded EEG data for 10 min while the patients kept their eyes closed, and the analysis was conducted using the Structural Brain Injury Index (SBII) algorithm. The SBII showed a sensitivity of 94.1% for detecting haemorrhagic strokes, with a specificity of 50.4% for stroke mimics serving as controls.

Another study assessed the Quick-20 EEG system for diagnosing patients with acute stroke.[46] Out of the 100 patients involved, 63 were diagnosed with acute stroke/TIA, among whom 7 patients had ICH. EEG signals alone achieved a sensitivity of 65% and specificity of 80%, with an AUC of 78.2 for detecting acute stroke/TIA. Combining EEG with clinical data using deep learning improved diagnostic performance, yielding a sensitivity of 79%, a specificity of 80% and an AUC of 87.8.

## DISCUSSION

Overall, we identified 15 studies that reported early clinical features of ICH among suspected stroke patients and 7 studies that tested portable technologies to detect ICH and other conditions.

### Early clinical features of ICH in suspected stroke patients

Diagnosing ICH based solely on clinical features can be difficult, as many features of ICH overlap with other types of strokes.[47] This scoping review has identified potential clinical features that can help distinguish patients with ICH from those with other stroke types or non-stroke diagnoses. The combination of these features might be useful in improving the early prediction of ICH. However, it is important to note that the included studies investigated and reported different clinical features for patients with ICH across various stroke subtypes, and some studies also included TIA and stroke-mimicking conditions in their analyses. In addition, some studies only conducted univariable analysis, and caution is advised when interpreting our results as there could be bias in the studies, such as confounding. Heterogeneity in the definition and categorisation of variables was also observed. Owing to these methodological limitations, we recognise that conducting a meta-analysis of individual features would be highly challenging and unlikely to yield meaningful insights relevant to the aims of this review.

Given the considerable heterogeneity among the included studies, identifying consistent predictors for the diagnosis of ICH was challenging. In terms of patients'

general information and medical history, neither age nor sex consistently predicted the final diagnosis of ICH across the studies. However, certain factors were reported to be more strongly associated with the presence of ICH, including a history of HTN, prior ICH, haemodialysis, alcohol consumption and the use of warfarin. In contrast, patients with ICH were less frequently found to have a history of AF, CAD, dyslipidaemia, DM, PAD, prior TIA, or IS. If available in the prehospital setting, ECG monitoring can be used to assess the presence of AF, which may help differentiate ICH from IS.[26 29 48 49] Nevertheless, the associations between these factors and ICH varied across studies. These results contradict a previous meta-analysis[15] that found a greater likelihood of ICH in male patients aged 60 years or less, while a history of TIA, PAD and AF increased the probability of IS being diagnosed.

The initial signs and symptoms of ICH, such as head-ache, vomiting and decreased LOC, also demonstrated variability across the included studies. While these manifestations were frequently observed, some studies did not find them to be exclusively indicative of ICH. This discrepancy could potentially be attributed to the inclusion of patients with SAH in certain studies.[28 30] Patients with SAH typically present with a sudden onset of headache, vomiting, decreased LOC and neck stiffness.[50 51] Consequently, relying solely on these clinical features for the prediction of ICH may lack specificity when compared with patients with SAH.

The majority of studies in this review consistently reported that patients with ICH experienced more severe neurological deficits at onset compared with other subtypes of stroke, TIA and stroke-mimicking conditions. Furthermore, patients with ICH were less commonly observed to have neurological deficits on awakening when compared with patients who had an IS. These findings are in line with other studies that have reported a higher frequency of hemiplegia/hemiparesis and sensory deficits in patients with cerebral haemorrhage,[48 50] while stroke symptoms on awakening were more frequently observed in patients who had an IS.[50 52]

To assess and quantify neurological deficits, more than half of the studies used the NIHSS, which consistently showed higher scores in patients with ICH. In fact, a recent study investigating the relationship between early NIHSS scores and ICH volume found that for every 10 mL increase in ICH volume, the NIHSS score increased by 4.5 points.[53] However, despite this, the NIHSS is not routinely used by prehospital personnel for stroke assessment due to its complexity and time-consuming nature.[54] In an effort to address these limitations, a recent study demonstrated that the NIHSS can be an accurate and time-efficient tool for paramedics in the field,[55] suggesting the potential of using the NIHSS items to construct a prehospital prediction model for ICH.[29]

Several prediction models have been proposed to differentiate ICH from other suspected stroke patients in the prehospital setting.[25 27–30] These tools can assist prehospital personnel in making accurate and timely decisions for patients who had a stroke. However, their predictive abilities have not been thoroughly studied, compared and evaluated in an independent study. Therefore, further research is necessary to assess their performance, reliability and potential limitations or biases that may impact their effectiveness in real-world settings. The addition of the clinical features found in this review could further improve the performance of these tools.

Another important finding of this study is that patients with ICH had the highest BP parameters among all suspected stroke patients. Notably, there is a strong correlation between elevated SBP of ≥160 mm Hg and the initial haematoma volume in patients with ICH.[56] A post hoc analysis of the FAST-MAG trial revealed a distinct association between SBP and ultra-early neurological deterioration (defined as a deterioration of 2 or more points on the Glasgow Coma Scale within 2 hours of stroke onset) among patients with ICH.[57] According to the study, neurological deterioration in patients with ICH was associated with higher SBP on arrival at the ED. Additionally, imaging performed on arrival showed that patients with ICH with neurological deterioration had much larger haematoma volumes, suggesting that early HE may contribute to the deterioration.[57] Previous studies have also found an association between prehospital BP values (SBP and DBP) and the occurrence of neurological deterioration in ICH.[58 59] These findings partially align with our results, in which there was a very powerful relationship between increased prehospital SBP and the occurrence of ICH among patients who had a stroke with impaired LOC.[32]

Other vital signs, such as HR and $SPO_2$, were compared in some of the included studies and seemed to be insignificant in predicting the presence or absence of ICH. However, associations appear evident between cardiac arrhythmias in patients who had an IS or patients with LVO, while higher blood glucose levels were reported in patients with ICH on hospital admission.

### Portable technologies for ICH detection and differentiation
Numerous portable technologies have been designed to identify and distinguish ICH from other stroke subtypes or stroke-mimicking diagnoses. In addition to clinical features, this scoping review has identified portable devices that have the potential to improve the detection of ICH and could be used in various settings, including prehospital care.

In recent years, significant progress has been made in the prehospital diagnosis and treatment of patients with ICH with the introduction of mobile stroke units (MSUs).[2 60 61] However, due to the high costs associated with this technology, it is unlikely to become widely available for stroke care in the near future.[62] Thus, portable devices may serve as cost-effective alternatives to MSUs in the prehospital setting.

Some of the technologies evaluated showed greater accuracy than clinical assessments.[41 44 46] Antipova et al[44] developed a triage model that combined transcranial

sonographic findings with clinical assessment, resulting in significantly improved diagnostic sensitivity for detecting ICH. Hence, the combination of clinical features with the findings of these systems might further enhance the diagnostic accuracy of ICH.

The reviewed studies also highlighted the potential benefits of using these technologies for rapid diagnosis in the prehospital setting. For instance, in the VITAL study,[41] the VIPS technology demonstrated an average detection time of 30 s for severe strokes. Other technologies evaluated had median application times ranging from 10 min to 20 min (online supplemental table 2), depending on the specific technology employed. Shortening the time-to-treatment duration is crucial in stroke management to improve patient outcomes.[63] Therefore, it is important to ensure that the use of diagnostic technologies does not cause delays in transporting patients to definitive care.

A number of limitations were identified for each technology. Most notably, the discriminatory ability to differentiate ICH from other diseases was not evaluated with VIPS and portable EEGs.[41 45 46] Conversely, microwave and ultrasound imaging techniques yielded good results in differentiating between stroke subtypes.[40 42–44] However, the diagnostic accuracy of microwave technology to discriminate ICH from IS could be decreased with the development of cerebral oedema after IS.[40] On the other hand, transcranial ultrasound cannot produce useful images unless adequate temporal bone windows are available.[42–44] Furthermore, with this technology, pathological conditions such as haemorrhagic transformation of infarcts and gliomas could be misdiagnosed as ICH.[42 43]

While several diagnostic modalities discussed in this review show promise for prehospital stroke care, their diagnostic accuracy requires further improvement and validation using a large, diverse cohort of suspected stroke patients. Such validation is crucial to establish their clinical usefulness in the prehospital setting, where both sensitivity and specificity are important in determining appropriate management. Given the significant differences in treatment strategies between ICH and IS, it is imperative that the detection technologies can effectively differentiate between the two with high sensitivity and specificity. Reliable differentiation can enable the prompt initiation of appropriate treatment measures for ICH, including BP reduction and anticoagulation reversal, and facilitate triaging these patients to neurosurgical-capable facilities. Moreover, consideration should be given to the level of training required for personnel to competently operate these technologies.[24] For instance, devices such as transcranial ultrasound necessitate highly trained operators, which may limit their feasibility in the prehospital setting.

### Other novel approaches to detect and distinguish ICH

There are other novel technologies that exist for detecting and distinguishing ICH, but these fall beyond the scope of this review, either due to the technologies not being investigated in suspected stroke cases or their

lack of portability. One of the most studied among them is near-infrared spectroscopy (NIRS). In a pilot study, the handheld NIRS Infrascanner was evaluated for its ability to detect intracranial haematomas in 35 patients with traumatic brain injuries.[64] When compared with CT scans, which were considered the gold standard, the Infrascanner demonstrated a sensitivity of 88.89% (95% CI 50.7% to 99.4%) and specificity of 81.25% (95% CI 53.7% to 95.0%) in identifying intraparenchymal haematomas. However, its effectiveness in accurately distinguishing between stroke subtypes, particularly in identifying deep ICH, may be hindered by its limited detection range of approximately 3 cm beneath the skull.[64 65]

Besides brain diagnostic devices, the use of blood biomarkers is also a promising approach for the early detection and differentiation of stroke subtypes.[37 66] However, it remains unclear whether they can be used as a point-of-care diagnostic method in the prehospital setting.[67] This is currently being investigated in a prehospital trial (Biomarkers for Initiating Onsite and Faster Ambulance Stroke Therapies, Bio-FAST, ClinicalTrials. gov Identifier: NCT04612218) with point-of-care test devices.

### Limitations of the review

Our scoping review has some limitations that should be considered. Only one reviewer screened and analysed all of the data. Although a second reviewer independently screened 20% of the articles at the abstract, full-text and data extraction stages, there is a risk of review errors and bias. Additionally, the search was limited to English publications, which could have resulted in the exclusion of important studies published in other languages. Furthermore, some of the included studies examined the clinical features of ICH up to 24 hours of symptoms onset, which requires cautious interpretation of the findings regarding their external validity for prehospital evaluations. Lastly, owing to the nature of scoping reviews, the methodological quality of the studies included was not assessed.[20] This means that poor-quality studies were given equal weight as high-quality studies, which should be taken into account when interpreting the review findings.

### CONCLUSION AND FUTURE WORK

Our study identifies potential early clinical features of ICH that distinguish it from suspected stroke patients. However, the substantial heterogeneity observed in populations, clinical features and outcome variables among the included studies precludes the combination of these features in a meta-analysis. The variability in findings across studies highlights the need for further research and standardised tools to improve early prediction and differentiation of ICH. Future studies should focus on developing and validating predictive models or tools that incorporate easily obtainable clinical features for prehospital personnel. Additionally, it is important to consider an unselected cohort of suspected stroke patients in

future studies to ensure that the findings are more widely applicable and representative of real-world practice. Furthermore, large independent cohorts should be used to evaluate the performance, reliability and potential limitations of these predictive tools.

In addition, this study presents portable technologies for the detection of ICH and other stroke subtypes. While some of these technologies have shown promising results, the majority are in the early stages of development and require further improvement and validation in large cohorts of suspected stroke cases. Furthermore, the selected studies in this review reported different study designs, comparison groups and diagnostic aims, making it challenging to draw definitive conclusions about the superiority or preference for one technology over another. To better understand the potential benefits and feasibility of using portable devices for prehospital stroke care, future research should not only focus on reporting the diagnostic performance of the devices but also consider the practical aspects of their use in prehospital settings. This includes assessing the training requirements for device use, the time needed to acquire and interpret results and the ability of the technology to differentiate between stroke subtypes (ICH vs IS) and non-stroke diagnoses.

**Author affiliations**
[1]Division of Cardiovascular Sciences, The University of Manchester, Manchester, UK
[2]Department of Aviation and Marine, Prince Sultan Bin Abdulaziz College for Emergency Medical Services, King Saud University, Riyadh, Saudi Arabia
[3]Department of Emergency Medical Services, College of Applied Medical Sciences, Khamis Mushait Campus, King Khalid University, Abha, Saudi Arabia
[4]Manchester Centre for Clinical Neurosciences, Northern Care Alliance NHS Foundation Trust, Salford, UK
[5]Division of Informatics, Imaging and Data Science, The University of Manchester, Manchester, UK

**Correction notice** This article has been corrected since it was published. Licence updated to CC BY on 2nd August 2024.

**Acknowledgements** The authors wish to express their sincere gratitude to the peer reviewers of this study for their valuable input and suggestions.

**Contributors** MA is the guarantor of the content of the manuscript. All authors included in this paper fulfil the criteria of authorship. MA is the primary author of this work under the supervision of ARP-J and DJ. DJ conceptualised the idea for the review. MA and IA screened the studies and performed data extraction. MA conducted data analyses and wrote the manuscript with support from ARP-J and DJ. The manuscript was critically reviewed and approved by all authors for final submission.

**Funding** This study is part of MA's PhD research, which is funded by King Saud University, Riyadh, Saudi Arabia (grant no: NA), through the Saudi Arabian Cultural Bureau in the United Kingdom (grant no: NA).

**Competing interests** None declared.

**Patient and public involvement** Patients and/or the public were not involved in the design, or conduct, or reporting, or dissemination plans of this research.

**Patient consent for publication** Not applicable.

**Ethics approval** This is a scoping review; therefore, ethical approval is not required.

**Provenance and peer review** Not commissioned; externally peer reviewed.

**Data availability statement** All data relevant to the study are included in the article or uploaded as supplementary information.

**ORCID iDs**
Mohammed Almubayyidh http://orcid.org/0000-0003-3266-1236
David Jenkins http://orcid.org/0000-0001-6687-3507

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
