## [Reviewer comments · BMJ Open]

ARTICLE DETAILS

TITLE (PROVISIONAL)	Prehospital identification of intracerebral haemorrhage: a scoping review of early clinical features and portable devices
AUTHORS	Almubayyidh, Mohammed; Alghamdi, Ibrahim; Parry-Jones, Adrian; Jenkins, David

VERSION 1 – REVIEW

REVIEWER	Richards, Christopher T. University of Cincinnati
REVIEW RETURNED	22-Nov-2023

GENERAL COMMENTS	The authors present a high-quality scoping review of published studies that have attempted to identify common clinical features of patients diagnosed with ICH and studies that have evaluated portable diagnostics for ICH, with an overarching goal of improving prehospital detection of ICH. The authors found studies that report clinical features more commonly seen in patients with ICH, but heterogeneity in the studies' findings was evident. Identified studies show early promise for portable diagnostic devices, but accuracy is limited in current technology. (Introduction, Page 5, Lines 7-22.) To further emphasize the importance of this topic, the authors could consider adding the following concepts and references to the Introduction and/or Discussion: 1. In the 2022 American Heart Association/American Stroke Association ICH guidelines, there are several recommendations related to prehospital ICH care, specifically one recommendation regarding the importance of ICH screening (Section 3, Recommendation 2 in DOI: 10.1161/STR.000000000000407).2. The authors cite the FASTEST study protocol publication already in their Reference list. But the authors could highlight that highly reliable screens or diagnostics in the prehospital setting could allow for expanded access to clinical trials of therapeutics in the prehospital setting (FASTEST can enroll in the prehospital setting, but only in systems with mobile stroke units because of their CT scanners).3. The RACECAT study (doi:10.1001/jama.2022.4404) looked at the effect of a longer transport to a higher-level stroke center for patients with severe stroke and found that there was a signal toward worse outcomes in the subset of patients with ICH transported to a more distant site (DOI: 10.1001/jamaneurol.2023.2754). This was a subgroup analysis, but it highlights that the prehospital transport destination decision is another important outcome of improved prehospital differentiation of ICH.4. The differentiation between ICH and LVO carries significant importance as well. As the authors mention, current severe stroke
--

screens have difficulty discriminating between LVO ischemic stroke and ICH. However, the types of stroke centers that EMS may transport to are differently scoped – comprehensive stroke centers are certified for ICH and LVO, and thrombectomy-capable stroke centers (increasingly common) are certified for LVO but not necessarily ICH. This leads to either most severe strokes transported to CSCs, or ICH patients being transported to TSCs where neurosurgical and neuro-ICU capabilities may not be as robust. This recent publication has a sizable discussion on this topic and could be cited: doi.org/10.1161/STROKEAHA.123.039792.

(Methods, Page 6, Line 40.) The authors may wish to include in the Limitations that including studies that looked at clinical features of ICH up to 24 hours may have external validity limitations for prehospital evaluation of ICH, particularly because the authors report that haematoma expansion happens most frequently in the first 2-3 hours of onset (Introduction, Page 5, Lines 16-18).

(Methods, Page 7, Line 12.) Please specify if included primary research studies needed to be peer-reviewed to meet inclusion criteria.

(Results, Page 9, Lines 3-5.) If these three groups of clinical features were determined a priori, then please move this to the Methods. If these three groups were identified through the course of analysis, please indicate that these grouping emerged during the analysis (and therefore keep as a Result).

(Results, Page 8, Lines 24-32; Results, Page 9, Lines 6-8; Discussion, Page 17, Lines 11-17.) These various lines read more like results of an analysis, rather than a description of studies and the studies' results. This is subtle, but important. The current phrasing (e.g., "ICH were found to be younger") implies that a test of significance was performed in this analysis, which it was not (this verbiage almost implies a meta-analysis of the studies). Please replace with clearer language (e.g., "most studies found that patients with ICH were younger than..."). This clearer language is used more commonly in the Results and Discussion, but in the lines listed, it is not.

(Results, Page 10, Line 40.) For these two pilot studies, please include a description of the time from last known well to device application, if reported in the original studies. If not, include that as a limitation.

(Results, Page 12, Lines 35-37; Table 3, Citations 42-44.) Please clarify why TBI patients were included in the scoping review in the Methods (or present these three studies in a clearer/different way, as recommended below). The Introduction and Methods describe spontaneous ICH and set up comparisons in how clinical scales and portable technology can differentiate between spontaneous ICH and other acute intracranial processes. Therefore, the inclusion of TBI studies seems out of scope. I also think that including TBI in the clinical features set of articles would expand the list of included studies significantly (and again would be out of scope for this paper). Therefore, I would recommend removing these three citations from the analysis. That said, the NIRS technology is important to include. I would recommend moving this section near the Discussion section about biomarkers and labeling this section "Future Directions" or similar terminology.

	(Discussion, Page 14, Line 17.) The authors do mention the heterogeneity of included studies as well as the challenge of identifying a true signal among included studies because of the variable results of studies. The first lines of the Conclusions address this (as does the second paragraph in the Discussion), but I would recommend adding stronger language here at the beginning of this paragraph where the authors attempt to find some unifying signals in the heterogenous studies. Language here could also reflect these variable findings, using terms such as “preponderance of studies found...,” etc. (Discussion, Page 15, Lines 22-24.) The authors state that a prehospital NIHSS could help identification of ICH, but I’m not sure how this would differentiate between ICH and IS (including LVO), which is the tone of most of the paper. Perhaps consider tempering this line. (Discussion, Page 18, Lines 7-15.) While the section on biomarkers is interesting, it’s out of scope for the main results. That said (related to the TBI comment above), I would consider setting this section off as a “Future Direction” or “Other Considerations” section, highlighting that a true analysis of biomarkers was not part of this review, but is an important part of future innovations for detection of spontaneous ICH (as is NIRS). (Discussion, Page 18, Line 38.) There is little discussion about the potential for a current or future meta-analysis, given that the “study aims to assess the feasibility of conducting a future meta-analysis.” The results demonstrate significant heterogeneity in studies, and I agree with the authors that this limits “the ability to perform a meta-analysis of individual features.” However, if determining this was an aim of the paper, more attention needs to be dedicated to this aim. Therefore, I would recommend one of two paths: a) De-emphasize (i.e., remove) the aim of determining the feasibility of meta-analysis and add a line or two in the discussion about how a meta-analysis would be very limited given the heterogeneity of studies, or b) Add dedicated sections in the Methods, Results, and Discussion about how suitability of identified studies for a future meta-analysis was systematically determined.
--	---

REVIEWER	Ford, Gary University of Oxford Radcliffe Department of Medicine
REVIEW RETURNED	05-Dec-2023

GENERAL COMMENTS	This is a scoping review of early clinical features and portable devices for the pre-hospital detection of ICH. The search identified 25 studies – 10 reporting early clinical features, 10 portable devices. Findings were ICH patients had more severe neurological deficits and higher BP. Five technologies were identified with microwave and ultrasound imaging showing promise in distinguishing ICH from other diagnoses. The authors conclude there are potential clinical features for identification of ICH and that devices need further testing to assess their ability to differentiate ICH from ischaemic stroke and non-stroke diagnoses. 1. Differentiation of ICH from ischaemic stroke and stroke mimics in the pre-hospital setting using clinical features or technology could potentially be of some value, but it is extremely unlikely clinical features and portable technology (i.e. not CT brain imaging) will be able to perform well enough to enable IV thrombolysis to be
--

	administered in the field without brain imaging. However the identification of ischaemic stroke due to large vessel occlusion is a more important requirement for health systems, as that would enable more effective identification of patients who require transport to hospitals able to deliver thrombectomy. There is currently no evidence to support transfer of ICH outcomes are improved by transfer to comprehensive stroke centres rather than the nearest hospital with a hyperacute stroke unit. 2. The authors state this is the first review focused on identifying ICH in the pre-hospital setting but this is incorrect. See Chennareddy, S., Kalagara, R., Smith, C. et al. Portable stroke detection devices: a systematic scoping review of prehospital applications. BMC Emerg Med 22, 111 (2022) which identified 16 studies. The current review adds no new observations to the conclusions of that review with respect to technologies. 3. The observations on clinical features associated with ICH confirm findings from hospital and population studies that more severe deficits, higher BP, impaired conscious level, seizures and vomiting were found to be more common in most studies bur are weak differentiating symptoms for ischaemic stroke. The authors do not make reference to the may previous hospital studies which have compared presenting features in ICH and ischaemic stroke. The findings across the studies might be better represented in tabular form. 4. It is surprising no studies reviewed reported stroke deterioration / progression in the pre-hospital setting as a more common feature in ICH, although reference is made to this in the discussion in relation to the FAST-MAG study.
--	---

VERSION 1 – AUTHOR RESPONSE

Reviewer #1:

C1: (Introduction, Page 5, Lines 7-22.) To further emphasize the importance of this topic, the authors could consider adding the following concepts and references to the Introduction and/or Discussion: In the 2022 American Heart Association/American Stroke Association ICH guidelines, there are several recommendations related to prehospital ICH care, specifically one recommendation regarding the importance of ICH screening (Section 3, Recommendation 2 in DOI: 10.1161/STR.000000000000407).

R1: We thank the reviewer for this insightful comment. The suggested American Heart Association/American Stroke Association recommendation was discussed in our previously published study protocol (doi: 10.1136/bmjopen-2022-070228). However, to further emphasise the importance of this recommendation, we have added the following to our Introduction: “Recognising the importance of timely and accurate identification of ICH, the American Heart Association/American Stroke Association’s recent guidelines for the management of patients with ICH have highlighted the need to improve the prehospital recognition of ICH [9]”.

C2: The authors cite the FASTEST study protocol publication already in their Reference list. But the authors could highlight that highly reliable screens or diagnostics in the prehospital setting could allow for expanded access to clinical trials of therapeutics in the prehospital setting (FASTEST can enroll in the prehospital setting, but only in systems with mobile stroke units because of their CT scanners).

R2: We thank the reviewer for this important suggestion and agree that the ability to diagnose ICH easily and inexpensively in ambulances could facilitate the expansion of prehospital therapeutic trials.

We have now added the following text to the Introduction to highlight the importance of this point: “treatment strategies are currently under investigation in prehospital settings [6-8]. Facilitating the diagnosis of ICH in ambulances through simple and cost-effective methods could support the broader implementation of such therapeutic trials in the future”.

C3: The RACECAT study (doi:10.1001/jama.2022.4404) looked at the effect of a longer transport to a higher-level stroke center for patients with severe stroke and found that there was a signal toward worse outcomes in the subset of patients with ICH transported to a more distant site (DOI: 10.1001/jamaneurol.2023.2754). This was a sub-group analysis, but it highlights that the prehospital transport destination decision is another important outcome of improved prehospital differentiation of ICH.

R3: We thank the reviewer for highlighting the findings of the RACECAT trial, which allow us to further clarify the importance of distinguishing ICH in the prehospital setting. We have now cited the suggested reference and added the following to the Introduction section: “The recent RACECAT trial further emphasises the significance of prehospital decisions in stroke management, which found that bypassing the closest stroke centre for direct transport to a thrombectomy-capable centre was associated with worse outcomes for patients with ICH [10]”.

C4: The differentiation between ICH and LVO carries significant importance as well. As the authors mention, current severe stroke screens have difficulty discriminating between LVO ischemic stroke and ICH. However, the types of stroke centers that EMS may transport to are differently scoped – comprehensive stroke centers are certified for ICH and LVO, and thrombectomy-capable stroke centers (increasingly common) are certified for LVO but not necessarily ICH. This leads to either most severe strokes transported to CSCs, or ICH patients being transported to TSCs where neurosurgical and neuro-ICU capabilities may not be as robust. This recent publication has a sizable discussion on this topic and could be cited: doi.org/10.1161/STROKEAHA.123.039792.

R4: We thank the reviewer for raising this very interesting point. Indeed, it would be better to distinguish LVO from ICH, as patients with LVO may benefit from direct transportation to an endovascular-capable centre. We have addressed this point in the Introduction section of the revised manuscript and have cited the suggested reference as well.

C5: (Methods, Page 6, Line 40.) The authors may wish to include in the Limitations that including studies that looked at clinical features of ICH up to 24 hours may have external validity limitations for prehospital evaluation of ICH, particularly because the authors report that haematoma expansion happens most frequently in the first 2-3 hours of onset (Introduction, Page 5, Lines 16-18).

R5: Thank you for this excellent observation. We have now included the following statement in the Limitations section of the revised manuscript: “Furthermore, some of the included studies examined the clinical features of ICH up to 24 hours of symptoms onset, which requires cautious interpretation of the findings regarding their external validity for prehospital evaluations”.

C6: (Methods, Page 7, Line 12.) Please specify if included primary research studies needed to be peer-reviewed to meet inclusion criteria.

R6: Thank you very much for this comment. We have amended the Methods section in the revised manuscript to explicitly state that only peer-reviewed primary research studies published in English were considered for inclusion.

C7: (Results, Page 9, Lines 3-5.) If these three groups of clinical features were determined a priori, then please move this to the Methods. If these three groups were identified through the course of analysis, please indicate that these grouping emerged during the analysis (and therefore keep as a Result).

R7: Thank you very much for your insightful comment. We wish to clarify that the arrangement of clinical features into the specified groups was not predetermined but emerged during the analytical process. As you recommended, we have amended the manuscript to include this information in the Results section.

C8: (Results, Page 8, Lines 24-32; Results, Page 9, Lines 6-8; Discussion, Page 17, Lines 11-17.) These various lines read more like results of an analysis, rather than a description of studies and the studies' results. This is subtle, but important. The current phrasing (e.g., "ICH were found to be younger") implies that a test of significance was performed in this analysis, which it was not (this verbiage almost implies a meta-analysis of the studies). Please replace with clearer language (e.g., "most studies found that patients with ICH were younger than..."). This clearer language is used more commonly in the Results and Discussion, but in the lines listed, it is not.

R8: Thank you very much for your comment. We appreciate your feedback and agree that the language used in the specified lines could suggest a level of statistical analysis not present in our study. To address this concern, we have carefully revised the manuscript to clarify the presentation of our findings, especially in the listed lines.

C9: (Results, Page 10, Line 40.) For these two pilot studies, please include a description of the time from last known well to device application, if reported in the original studies. If not, include that as a limitation.

R9: We thank the reviewer for this suggestion. We have now included the time from last known well to device application in the revised manuscript, as reported in the original research.

C10: (Results, Page 12, Lines 35-37; Table 3, Citations 42-44.) Please clarify why TBI patients were included in the scoping review in the Methods (or present these three studies in a clearer/different way, as recommended below). The Introduction and Methods describe spontaneous ICH and set up comparisons in how clinical scales and portable technology can differentiate between spontaneous ICH and other acute intracranial processes. Therefore, the inclusion of TBI studies seems out of scope. I also think that including TBI in the clinical features set of articles would expand the list of included studies significantly (and again would be out of scope for this paper). Therefore, I would recommend removing these three citations from the analysis. That said, the NIRS technology is important to include. I would recommend moving this section near the Discussion section about biomarkers and labeling this section "Future Directions" or similar terminology.

R10: Thank you very much for your comment. Following your suggestion, we have removed the three citations related to TBI from our analysis to focus solely on studies of spontaneous ICH. Furthermore, we have added a new section titled "Other novel approaches to detect and distinguish ICH" dedicated to discussing other potential options for diagnosing ICH in the prehospital setting.

C11: (Discussion, Page 14, Line 17.) The authors do mention the heterogeneity of included studies as well as the challenge of identifying a true signal among included studies because of the variable results of studies. The first lines of the Conclusions address this (as does the second paragraph in the Discussion), but I would recommend adding stronger language here at the beginning of this paragraph where the authors attempt to find some unifying signals in the heterogenous studies. Language here could also reflect these variable findings, using terms such as "preponderance of studies found..." etc.

R11: Thank you very much for your feedback. We appreciate your input in enhancing the clarity of our discussion. As suggested, we have strengthened the language in the beginning of the paragraph to better reflect the variable findings between the studies.

C12: (Discussion, Page 15, Lines 22-24.) The authors state that a prehospital NIHSS could help identification of ICH, but I'm not sure how this would differentiate between ICH and IS (including LVO), which is the tone of most of the paper. Perhaps consider tempering this line.

R12: Thank you very much for your feedback. As suggested, we have revised the statement to more clearly indicate that the NIHSS variables could potentially be used to develop a predictive model for ICH in the prehospital setting.

C13: (Discussion, Page 18, Lines 7-15.) While the section on biomarkers is interesting, it's out of scope for the main results. That said (related to the TBI comment above), I would consider setting this

section off as a "Future Direction" or "Other Considerations" section, highlighting that a true analysis of biomarkers was not part of this review, but is an important part of future innovations for detection of spontaneous ICH (as is NIRS).

R13: Thank you very much for your comment. As mentioned in R10, we have added a new section titled "Other novel approaches to detect and distinguish ICH" to discuss other novel technologies for detecting and distinguishing ICH in the prehospital setting, including NIRS and blood biomarkers.

C14: (Discussion, Page 18, Line 38.) There is little discussion about the potential for a current or future meta-analysis, given that the "study aims to assess the feasibility of conducting a future meta-analysis." The results demonstrate significant heterogeneity in studies, and I agree with the authors that this limits "the ability to perform a meta-analysis of individual features." However, if determining this was an aim of the paper, more attention needs to be dedicated to this aim. Therefore, I would recommend one of two paths: a) De-emphasize (i.e., remove) the aim of determining the feasibility of meta-analysis and add a line or two in the discussion about how a meta-analysis would be very limited given the heterogeneity of studies, or b) Add dedicated sections in the Methods, Results, and Discussion about how suitability of identified studies for a future meta-analysis was systematically determined.

R14: Thank you very much for the constructive feedback. We have now removed the aim of determining the feasibility of a future meta-analysis, and we have clarified in the Discussion that, due to significant methodological variations between the included studies, "we recognise that conducting a meta-analysis of individual features would be highly challenging and unlikely to yield meaningful insights relevant to the aims of this review".

Reviewer #2:

C1: Differentiation of ICH from ischaemic stroke and stroke mimics in the pre-hospital setting using clinical features or technology could potentially be of some value, but it is extremely unlikely clinical features and portable technology (i.e. not CT brain imaging) will be able to perform well enough to enable IV thrombolysis to be administered in the field without brain imaging. However the identification of ischaemic stroke due to large vessel occlusion is a more important requirement for health systems, as that would enable more effective identification of patients who require transport to hospitals able to deliver thrombectomy. There is currently no evidence to support transfer of ICH outcomes are improved by transfer to comprehensive stroke centres rather than the nearest hospital with a hyperacute stroke unit.

R1: Thank you very much for your comment. We agree that identifying LVO is an emerging priority in the prehospital setting. However, the importance of identifying patients with ICH in the prehospital has been further emphasised in the Introduction section of the revised manuscript, following the recommendations of Reviewer #1 (C1–C4). In addition, we acknowledge that strong evidence regarding the optimal transport destinations for patients with ICH is currently lacking, but as discussed in R3 above, the RACECAT trial found that "bypassing the closest stroke centre for direct transport to a thrombectomy-capable centre was associated with worse outcomes for patients with ICH"; this approach, however, could be more beneficial for patients with LVO. Hence, understanding the optimal destination for suspected stroke patients requires the implementation of differential destination protocols, which can be facilitated by the ability to distinguish ICH from other diagnoses, especially from LVO, in the prehospital setting (Richards et al., Stroke 2023).

C2: The authors state this is the first review focused on identifying ICH in the pre-hospital setting but this is incorrect. See Chennareddy, S., Kalagara, R., Smith, C. et al. Portable stroke detection devices: a systematic scoping review of prehospital applications. BMC Emerg Med 22, 111 (2022) which identified 16 studies. The current review adds no new observations to the conclusions of that review with respect to technologies.

R2: Thank you for your comment. After addressing C10 above, this review is, to the best of our knowledge, the first to investigate the possibility of identifying spontaneous ICH in the prehospital setting. The study by Chennareddy et al., 2022 (BMC Emergency Medicine), did identify 16 studies; however, their focus was on the ability of portable technologies to predict stroke in general, rather than specifically detecting or distinguishing spontaneous ICH in suspected stroke cases. Moreover,

some of the studies they reviewed did not include ICH in their analysis, and others included children and cases of traumatic ICH, which are not relevant to our current objectives. Given these distinctions, our study contributes to the literature by advancing knowledge about the ability of portable technologies to detect and differentiate spontaneous ICH in adult patients with suspected stroke.

C3: The observations on clinical features associated with ICH confirm findings from hospital and population studies that more severe deficits, higher BP, impaired conscious level, seizures and vomiting were found to be more common in most studies but are weak differentiating symptoms for ischaemic stroke. The authors do not make reference to the many previous hospital studies which have compared presenting features in ICH and ischaemic stroke. The findings across the studies might be better represented in tabular form.

R3: Thank you for this excellent observation. We would like to clarify that in the Discussion section, we carefully compared the findings on the clinical features of ICH with those from similar or closely related studies, particularly those investigating stroke subtypes and non-stroke diagnoses (e.g., stroke mimics) to ensure a thorough understanding and identification of clinical features critical for distinguishing ICH. As suggested, we have included a table in the revised manuscript (Table 2) to summarise the identified clinical features for distinguishing ICH from other conditions.

C4: It is surprising no studies reviewed reported stroke deterioration / progression in the pre-hospital setting as a more common feature in ICH, although reference is made to this in the discussion in relation to the FAST-MAG study.

R4: Thank you for the insightful comment. Indeed, neurological deterioration is a common complication early after ICH; however, none of the included studies provided a definition for deterioration or progression of symptoms or found it to be specific to ICH. On the other hand, the reference to the exploratory analysis of the FAST-MAG trial aimed to elucidate one of the findings regarding the strong relationship between increased SBP and the occurrence of ICH among stroke patients with impaired LOC.

VERSION 2 – REVIEW

REVIEWER	Richards, Christopher T. University of Cincinnati
REVIEW RETURNED	21-Mar-2024

GENERAL COMMENTS	Thank you for your thoughtful consideration of my previous comments. I believe that my concerns have been adequately addressed by the revisions.
--

REVIEWER	Ford, Gary University of Oxford Radcliffe Department of Medicine
REVIEW RETURNED	05-Mar-2024

GENERAL COMMENTS	The authors have addressed my previous comments. My one comment is that on the basis of the data presented the conclusion drawn by the authors 'While these devices have shown promise in detecting ICH, further testing is required to assess their ability to differentiate between stroke subtypes (ICH vs ischaemic stroke) and non-stroke diagnoses' seems overly optimistic. .To date none of the technologies or clinical assessments provide adequate discrimination so further testing will not change this conclusion. What is needed is further development of the technologies. The authors may wish to consider amending their conclusion to reflect
---

	this.
--	-------

VERSION 2 – AUTHOR RESPONSE

Reviewer #1:

C: Thank you for your thoughtful consideration of my previous comments. I believe that my concerns have been adequately addressed by the revisions.

R: We sincerely thank Reviewer #1 for reading the revised manuscript and appreciate this feedback.

Reviewer #2:

C: The authors have addressed my previous comments. My one comment is that on the basis of the data presented the conclusion drawn by the authors 'While these devices have shown promise in detecting ICH, further testing is required to assess their ability to differentiate between stroke subtypes (ICH vs ischaemic stroke) and non-stroke diagnoses' seems overly optimistic. To date none of the technologies or clinical assessments provide adequate discrimination so further testing will not change this conclusion. What is needed is further development of the technologies. The authors may wish to consider amending their conclusion to reflect this.

R: We sincerely thank Reviewer #2 for the positive feedback and for this important comment to improve the manuscript. As per your suggestion, we have amended the statement in the Conclusion to now read: "While these devices have shown promise in detecting ICH, further technological development is required to distinguish between stroke subtypes (ICH vs ischaemic stroke) and non-stroke diagnoses".